# Changes in the Infant Mortality Rate in Twin Towns of Brazil: An Ecological Study

**DOI:** 10.3390/children9111662

**Published:** 2022-10-30

**Authors:** Heluza De Oliveira, Eliana Wendland

**Affiliations:** Post-Graduate Program in Pediatrics, Universidade Federal de Ciências da Saúde de Porto Alegre, Porto Alegre 90050-170, Brazil

**Keywords:** infant mortality, border health, ecological studies

## Abstract

Background: The infant mortality rate (IMR) is a proxy of the living and health conditions of a given population, which allows us to assess the risk of death for children under one year. Although there is, in general, a reduction in infant mortality in Brazil little is known about this indicator in the regions and cities located on the international borders of the Brazilian territory and the changes that occurred in the face of the migratory impact of the Americas in the period from 1996 to 2020. The objectives of this study are to assess IMR in Brazilian Twin Cities (municipalities that are located on the border with a large influx of people) and its social determinants over time. Methods: This is an ecological study, whose units of analysis were the Brazilian Twin Cities, between 1996 and 2020, based on data on births and deaths in children under one year, available in the public vital information system in Brazil. Data were identified by the city in which the infant death occurred in addition to the mother’s primary city of residence. Correlation measurements were performed to test the associations of the IMR means between the independent variables. Results: The Twin Cities (Bonfim, Tabatinga, Pacaraima, Porto Murtinho, Cáceres, Foz do Iguaçu, Santo Antônio do Sudoeste e Dionísio Cerqueira) had higher numbers of infant deaths per place of occurrence than the number of deaths per place of maternal residence. The Northern Twin Cities exhibited the highest IMRs. Cities in the Midwest region showed variability. In the South region, most cities showed low rates. A positive correlation was identified with the Gini index with r = 0.67 and a negative correlation with the Municipal Human Development Index indicator of r= −0.70. Conclusions: The averages of IMRs in the Twin Cities were higher than in their States. In recent years, there has been an upward trend in infant mortality in these cities.

## 1. Introduction

The Infant Mortality Rate (IMR) is an important indicator of social well-being, which estimates the risk of death for live births (LB) during their first year of life. It is directly related to the socioeconomic and sanitary conditions of the population. These early deaths could be considered avoidable [1,2].

The magnitude of the IMR has been associated with the characteristics of the economic development model of a country or region [3,4]. Since the 1980s, the reduction in infant mortality has been a worldwide phenomenon [5]. Brazil, despite showing a significant decline, with (13.4 per thousand LB in 2018) [6] still has higher rates than several Latin American countries such as Argentina, Uruguay and Chile, which have IMRs of less than five deaths per thousand LB. In relation to developed countries, this difference is even more significant in France, Germany, Israel, Italy, Portugal and Japan, the IMRs have only four deaths per thousand LB [7].

Brazil has the largest public health system in South America and has an international border with nine countries, namely: French Guiana, Suriname, Venezuela, Colombia, Peru, Bolivia, Paraguay, Argentina and Uruguay [8], remaining subject to the migratory impact of these countries.

Among the Brazilian border cities, 33 are considered Twin Cities. They are those that are characterized by urban integration with neighboring countries, with high socioeconomic and cultural flows fundamentally from work, study, consumption and access to public services, especially health and education [9].

The recent wave of migration, seen in different parts of the world, has also occurred in Brazil, with a large influx of refugees already recognized in the country. Of these, 36% are Syrians; 15% Congolese; 9% Angolans; 7% Colombians and 3% Venezuelans [10,11]. The Twin Cities may be impacted on the health system by migrants from contiguous cities seeking care in Brazil or even functioning as temporary migration corridors. Thus, our objective is to evaluate the IMRs of the set of Brazilian Twin Cities, identifying trends over time.

## 2. Materials and Methods

This is an ecological study with time series analysis (1996–2020), using different vital public information systems in Brazil.

### 2.1. Study Population and Sample

The population of this study is formed by all records of LB and infant deaths, of children under one year of age, identified by place of occurrence and residence in the death information record, in the 33 Brazilian Twin Cities.

Twin Cities correspond to population densities cut by the borderline (land or river, articulated or not by infrastructure work). These cities have great economic and cultural potential. There is an intense level of movement of people, goods and capital. [12,13]. These cities are located in the states of Acre, Amazonas, Amapá, Mato Grosso do Sul, Paraná, Rondônia, Roraima, Rio Grande do Sul and Santa Catarina. In this way, Brazil shares its borders with Argentina, Paraguay, Uruguay, French Guiana, English Guiana, Bolivia, Colombia, Peru and Venezuela [14].

### 2.2. Criteria for Inclusion and Exclusion

Inclusion criteria were all registered infant deaths in children under one year of age, according to the place of occurrence and maternal residence, in the Brazilian Twin Cities, in the period 1996 to 2020. Data on infant deaths over one year of age were excluded of age, of non-border municipalities and of those border cities that are not considered Twin Cities.

### 2.3. Data Source

Data collection was performed through the DATASUS portal (www.datasus.gov.br) pdf (accessed on 29 April 2021)., using the Mortality Information System (SIM) available on the website (https://datasus.saude.gov.br/mortalidade-desde-1996-pela-cid-10) (accessed on 29 April 2021). and the Information System on LB (SINASC) through the address: https://datasus.saude.gov.br/nascidos-vivos-desde-1994 (accessed on 29 April 2021).

The SIM provides the number of deaths of children under 1 year of age and the calculation instrument (data entry) of this system is the Death Certificate (DC) [15]. The SINASC aims to gather information on births occurring throughout the national territory generated from the Birth Certificate (BC), standardized by the Ministry of Health [15,16]. The information collected in these systems is made available in TABWIN and TABNET^18^ formats.

Social indicators from the set of Twin Cities were also used, such as demographic data from 2010 and the population estimate for 2019 [17].

The Human Development Index (HDI-M) was extracted from the database of the Brazilian Institute of Geography and Statistics (IBGE) [18] and was categorized into development ranges as follows: “Very Low” (from 0 to 0.499), “Low” (from 0.500 to 0.599), “Medium” (from 0.600 to 0.699), “High” (from 0.700 to 0.799) and “Very High” (0.800 to 0.899) [19]. The Gini Index, which serves to identify the degree of concentration of income in a given group, revealing the difference between the income of the poorest and the richest, varies from zero to one. The value zero represents the situation of equality, that is, everyone has the same income. Value one is at the opposite extreme, that is, one person holds all the wealth [20].

The years 1991, 2000 and 2010 were used from the demographic censuses of the database of the Institute of Applied Economic Research (IPEA) [21]. Additionally, the percentage data of the Coverage of the Family Health Strategy Teams (FHST) were used. It was based on the national target parameter: high coverage (75 to 100%) medium coverage (50 to 74.9%) low coverage (0 to 49.9%), from the years 2007, 2010 and 2020, collected from the bank data from the Primary Health Care Secretariat (PHCS) [22]. The outcome variable (IMR) was calculated using the direct method, which consists of relating the total number of deaths in children under one year of age divided by the total number of LB per thousand [23]. All IMRs were aggregated between the years 1996 to 2020 and their averages were removed.

### 2.4. Statistical Analysis

The IMR was calculated from the data available in SIM and SINASC for the place of residence and occurrence using Excel 2010 Software for Windows. Data were tabulated separately by location: residence and occurrence. Data information was descriptively organized in tables and graphs in which the average IMR values of the cities were compared with the averages of the regions. According to the RIPSA [23] network, IMR can be classified as high, medium or low. The IMRs considered high are those equal to or greater than 50 deaths per thousand LB. Values between 20 and 49 per thousand LB are considered average. Additionally, low rates are those that have infant mortality of less than 20 per thousand LB. The association between the independent variables (Gini Index and HDI-M) and the dependent variable (the means of the IMRs) was estimated using the Pearson correlation test.

Ethical considerations: The study was carried out from a secondary source database, available for public research, whose information is aggregated, without the possibility of individual identification, and is therefore not subject to review by the CEP/CONEP system according to RESOLUTION No. 510, OF 07th APRIL 2016 [24].

## 3. Results

During the period 1996 to 2020, in Brazil, 491,761 births were registered by SINASC and 8360 deaths in children under one year of age were reported in SIM, in Twin Cities. Four cities in Rio Grande do Sul (Aceguá, Barra do Quaraí, Chuí and Porto Mauá) did not present data on the city in which the infant death occurred in addition to the mother’s primary city of residence.

While most of the Twin Cities in the North region had the HDI-M classified as medium to low, in the South most cities had HDI-M considered high. In the Center-West region, there was great variability between municipalities, and in the cities of Corumbá (0.700) and Ponta Porã (0.701), the HDI can be considered high. Bela Vista (0.698) and Mundo Novo (0.686) have medium indexes and Porto Murtinho (0.666), Paranhos (0.588) and Coronel Sapucaia (0.589) are low (Appendix A).

In general, social inequality is high in all cities observed, with a worsening of the Gini Index over the years, especially in cities located in the North and Central-West regions. However, the Twin Cities in the southern region of Brazil have shown, over the years, an improvement in the Gini index. For example, the Twin City Mundo Novo, located in the Midwest region, which in 1991 had a Gini value of 0.575, in 2010 managed to decrease it to 0.514 (Appendix A).

There is an inverse association between FHS coverage and the Gini index. The North region has the largest coverage by the ESF while the South region has the lowest coverage (Appendix A).

The North region had the highest IMRs both by occurrence and by residence. For example, we have the rates in the city of Epitaciolândia, which are quite high in the available years. There was an increase in IMR in two cities in the last period, such as Santa Rosa do Purusand Tabatinga. Some IMRs were classified as medium levels (Appendix A). The cities of Bonfim and Pacaraima, in 2020, had high IMRs per occurrence, respectively, 33.06 and 24.27 deaths/1000 LB (Figure 1).

In the Midwest region, there was, in general, a decrease in IMR over the years, with the exception of Paranhos and Mundo Novo. The latter showed an increase in IMR in the last period, from 18.79 to 21.39 deaths per 1000 LB (Appendix A). The South region also presented an absence of data, with the cities of Aceguá, Barra do Quaraí, Chuí and Porto Mauá not being tabulated as they did not have data available for calculating the IMR and only half of the cities had information on recent years.

Among the cities in the South region that presented available data, most had low IMRs, with the exception of the city of Quaraí, where mortality was higher, with the IMR per residence classified as an average with 21.67 deaths/1000 LB in 2020. Additionally, in the South region, the Twin Cities Foz do Iguaçu-PR, Dionísio Cerqueira and Santo Antônio do Sudoeste showed a stable trend of IMRs due to higher occurrences, being higher than the IMRs per residence (Figure 2).

In general, we can observe that there was a decrease in infant mortality. However, in the observed periods of 2014/2015 and 2016/2017, several cities showed an increase in IMRs. In Epitaciolândia—AC, despite an important reduction in the number of infant deaths over the years, the IMRs per residence increased from 2004 to 2005, from 13.38 per 1000 LB to 55.56 per 1000 LB, which meant an increase of 315.24%. In 2010, there was also an increase of 15.35%.

A similar pattern occurred in Pacaraima—RR from 2014 to 2015. There, there was an increase from 14.71 per 1000 LB to 16.62 per 1000 LB with a growth of 12.98%. This twin city was decreasing its number of infant deaths in recent years, however, there was a growth of 99.74% in records between the years 2016 and 2017.

The same situation can be seen in the city of Bonfim—RR between 2010 and 2011. Their IMR by maternal residence showed an increase of 673.20%. The IMRs due to the occurrence of Twin Cities such as Bela Vista—MS from 2000 to 2001 showed an increase of 29.57%. Between 2016 to 2017, the increase was 31.10%. In Mundo Novo—MS, between 2008 and 2009, it increased by 108.22%, while in Dionísio Cerqueira—SC, between 2010 and 2011, the increase was 29.76% (Figure 3).

There is a positive and statistically significant correlation (*p*-value < 0.001) between IMR and Gini index, with r = 0.67 (Figure 4). The Twin Cities that exhibited moderate IMR averages between 2010/2011 also showed high Gini indices, that is, closer to 1 (the maximum income inequality being). Cities in the North region stand out with the worst outcomes, such as the twin city of Santa Rosa do Purus—CA, with high income inequality close to 0.77 and an average IMR of 38.91 deaths/1000 LB. Next, in the Midwest region, we observe the city of Paranhos—MS with 31.26 deaths/1000 LB, and its Gini index was 0.65. It is identified that the higher the Gini index, the higher the mortality rate.

The HDI-M showed a negative correlation with the IMR (r= −0.70) and a statistically significant one (*p*-value < 0.001) (Figure 5). It is noteworthy that the higher the HDI-M, the lower the average of IMRs in the region, showing an inverse relationship between development and infant deaths. It is observed that cities in the North and Center-West regions have worse outcomes, such as the cities of Santa Rosa do Purus with Low Development with an HDI-M 0.517 and an average IMR of 38.91 deaths/1000 LB. In sequence, the Twin City Paranhos—MS with 31.26 deaths/1000 LB and the HDI-M 0.588, with low development.

## 4. Discussion

This is the first study that evaluated IMR in the Twin Cities, after the increase in the recent migratory flow in Brazil. The findings showed that, despite a decline in child mortality in the country, large disparities still exist. The Twin Cities of the North, South and Midwest regions, in 2018–2020, exceeded the infant death rates in their states, with the exception of Porto Murtinho in Mato Grosso do Sul higher than the number of infant deaths by place of maternal residence, indicating an influx of patients to these municipalities, even though these are not regional health care centers. Additionally, an inversion in the profile of the decreasing pattern of mortality was identified, with an increase in IMR in recent years.

Some limitations deserve to be pointed out. One of them concerns the use of secondary data, which suffer a delay in their registration with the system and are liable to being under-registered. Furthermore, it is an ecological study, allowing only inferences about populations, which cannot occur at the individual level. The lack of recent census data also impacts the estimates, since it involved the 2010 reference population.

Studies by Giovanella and Guimarães, (2007) [25]; Silva, (2014) [26] and Silva et al. (2017) [27] point to the occurrence of underreporting of records of health care provided in cross-border regions and correlate the increase in this underreporting due to the entry of migrants in search of informal care.

In Brazil, infant mortality remains a major public health problem, despite decreasing rates over the years. The IMRs of most of the Twin Cities remain higher than those of their states. All Twin Cities had an average IMR higher than the tolerable by the World Health Organization (WHO), which is 10 deaths per thousand LB [28], in the years 2016/2017.

However, the rates show inequalities even in those cities with lower IMRs in the South region, followed by the Midwest. This finding is similar to those reported by Peiter (2005) [29] and Souza and Silva (2018) [30] who claim that the lowest rates are concentrated in the Twin Cities in the South region and the highest rates in the North region. This inequality in the indicators has its origin in the differences in living conditions in these regions, resulting from urban infrastructure, income and employment, which vary substantially between North and South.

Another important aspect may be the cross-border dynamic, due to the economic integration project promoted by Mercosur^33^, which resulted in an improvement in the Gini index. The Twin Cities belonging to the North region, on the other hand, have more precarious living conditions and worse access to health services [29,30,31,32], with lower quality of prenatal care and less access to tertiary care services [33].

The main causes of death in children under one year of age in the municipalities belonging to the border zones were neonatal/perinatal causes resulting from poor care for pregnant women (22.8%) and preventable deaths due to lack of care for the newborn. (17.3%) [30]. Differences in health care follow the same pattern as infant mortality, with worse indicators in the North, an important variability in the Midwest, depending on the sub-region, and better indicators in the South [30]. Most cities located in the border area do not have quality care in tertiary care or secondary care services contributing to the increase in mortality [34].

An increased risk of death from preventable causes in children living in the Twin Cities had already been identified by Souza and Silva (2018) [30]. However, this study has not yet been able to detect the reversal observed in the decreasing pattern of mortality, with an increase in IMRs in recent years. Mortality increased by about 3% between 2015 and 2016, driven by the decrease in the number of LB in 2016, since pregnancies were avoided after the Zika virus epidemic and also by the increase in the number of infant deaths in the post-neonatal deaths, infant deaths from infectious causes.

The worsening of living conditions and the lack of access to medical care, which may be caused by cuts in social investments and in the SUS (Unified Health System) in recent years, were associated with an increase in mortality in specific categories [35,36]. Additionally, the impoverishment of the population and fiscal austerity measures, implemented since 2015, led to a growing reduction in spending on social welfare programs, cuts in health investment and reductions in income transfer programs such as the Bolsa Família Program. This had a direct impact on infant mortality in the country [36].

Some border municipalities had IMRs per place of occurrence greater than the number of deaths per place of maternal residence, indicating an influx of patients to these cities. Most of the Twin Cities are small and do not have much support in terms of health care.

The recent wave of migration observed in different parts of the world has also occurred in Brazil, with a large increase in refugees. In 2020, 82,520 permanent visas were requested, 65.1% of which were Venezuelans, followed by 20.1% Haitians and 4.8% Cubans [37]. The Twin Cities have been suffering an overload of assistance due to the search for care by foreigners in the Brazilian health system in neighboring cities that border or act as temporary migration corridors.

The study by Lima et al. (2020) [38] found that a large number of Venezuelan women crossed the border to seek care in maternity hospitals in Brazilian hospitals. In the cities of Roraima, Venezuelans are exposed to subhuman living conditions, with a lack of clean water and food, increasing the risk of infant deaths. 

Demographic, economic and cultural characteristics in border municipalities can directly lead to the probability that a child reaching his first year of life is greater than that of the group of non-border municipalities [39] Infant mortality is considered a proxy for development, as it is directly associated with indicators such as the Municipal Human Development Index (HDI-M) and the Gini Index [40] with inequalities in mortality being directly connected with differences in these indicators.

The pattern of increased mortality in recent years in the Twin Cities, as well as the increase identified in some specific categories, is also associated with worse living conditions in these cities. The HDI-M, which measures the well-being of a population, especially children, had lower averages in the Twin Cities (0.674) than the Brazilian average (0.727), with the exception of Foz do Iguaçu (0.751) [32].

These cities have deficiencies in education, health deficiencies, and worse employment and income, in general, when compared with other non-border Brazilian municipalities. Additionally, the border strip is marked by the contradiction between what is legal and what is illegal, attracting groups that use illegality to entice young people and women as mules of trafficking; men for the crossing of drugs, weapons, goods, etc. [41,42].

Social inequality, measured by the Gini index, is also an important factor for infant mortality, showing a direct correlation with this in the Twin Cities. The regions with the highest mortality are also those with the highest indicators. Municipalities in the South region, which in general have lower mortality rates, also have lower Gini indexes [40]. The Twin Cities in the South region have the greatest intensity of cross-border interactions and a better offer of services when compared with cities in the North and Midwest regions [40].

According to Guimarães et al. (2016) [41], the South of Brazil experiences post-modern inequality, marked by non-inclusive industrialization, prejudice and social separation, which may explain inequalities regarding the IMRs observed within this region. The North region, which has the highest mortality rates, also has the greatest inequality. The municipalities belonging to the Brazilian border strip present levels of social inequality above 50%, especially the municipalities of Roraima and Amazonas.

The care model focused on primary care and family health has been associated with the improvement of health indicators such as, for example, low birth weight and lower infant mortality) [43,44]. One of the SUS guidelines is the implementation of the Family Health Strategy (FHS) in priority populations that have social vulnerabilities, low education, and income, among others, leading to equity in health care [43].

The Twin Cities in the North region have the largest coverage of the FHS. Poorer regions such as the North and Northeast, which have the highest levels of poverty in the country, also have the greatest coverage [33]. The introduction of the FHS was later in the South region, where some Twin Cities did not have FHS coverage until mid-2006. All cities only reached implementation in 2020.

## 5. Conclusions

We show that most of the Twin Cities have high infant mortality rates, not only higher than recommended by the World Health Organization (WHO) but higher than the states where they are located. In addition, we identified that there is a change in the pattern of IMRs in these cities, which show an increasing trend of infant mortality in recent years in cities belonging to the North region, followed by the South region. Recent changes in the mortality pattern with mortality rates per occurrence greater than those of residents in border towns without a specialized assistance network may suggest a search for these towns by migrants. The hypothesis of an association between recent migratory flows and infant mortality needs to be explored through other study designs and deserves the attention of managers and researchers.

There is an important gap in information about infant mortality in the Twin Cities and the migrant population, as well as the impact on the health systems of these cities. This information gap also occurs in other Latin American countries. These cities have different characteristics that demand greater attention in terms of surveillance and health care, caused by the rapid and, at times, intense flow of people, and that need to be taken into account by health managers.

## Figures and Tables

**Figure 1 children-09-01662-f001:**
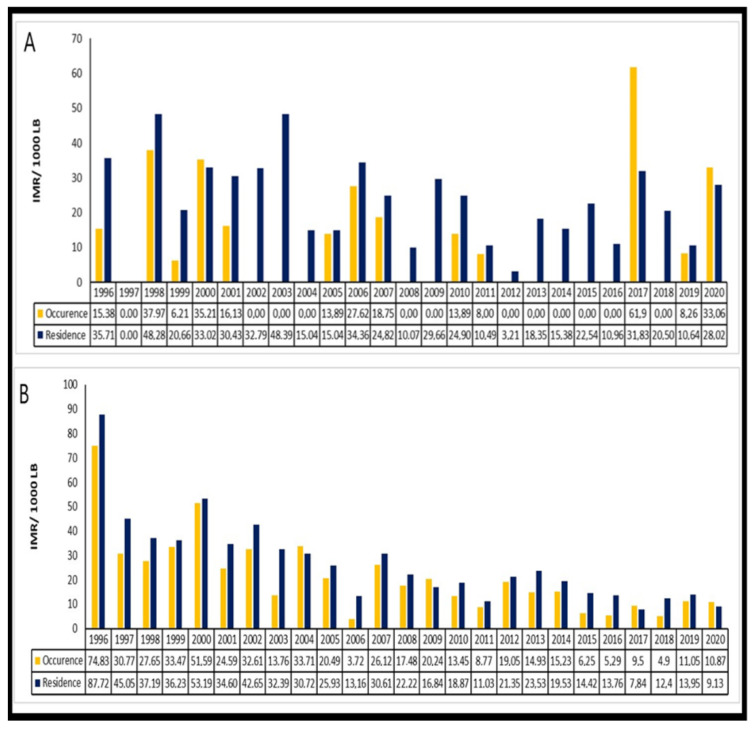
Comparison of infant mortality rate by occurrence and residence in the following selected cities in the North region: (**A**) Bonfim and (**B**) Pacaraima. Brazil, 1996–2020.

**Figure 2 children-09-01662-f002:**
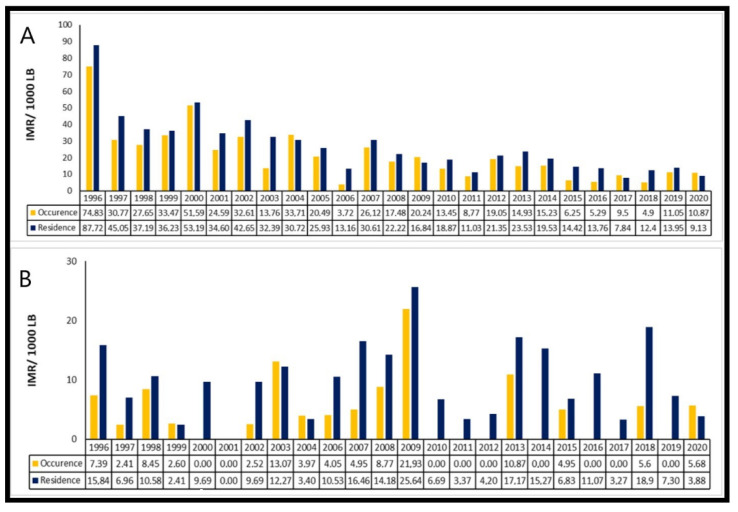
Comparison of infant mortality rate by occurrence and in the following selected cities in the South region: (**A**) Foz do Iguaçu and (**B**) Santo Antônio do Sudoeste. Brazil, 1996–2020.

**Figure 3 children-09-01662-f003:**
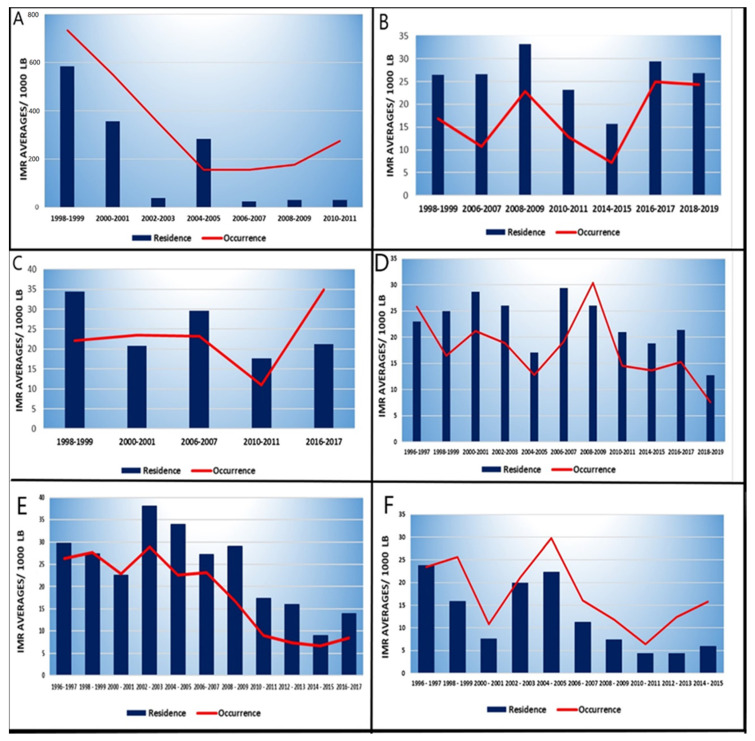
Change in infant mortality pattern through IMR averages per occurrence (red line) over the years in some Twin Cities: (**A**) Epitaciolândia–AC; (**B**) Pacaraima–RR; (**C**) Bonfim–RR; (**D**) New World–MS; (**E**) Bela Vista–MS and (**F**) Dionisio Cerqueira–SC.

**Figure 4 children-09-01662-f004:**
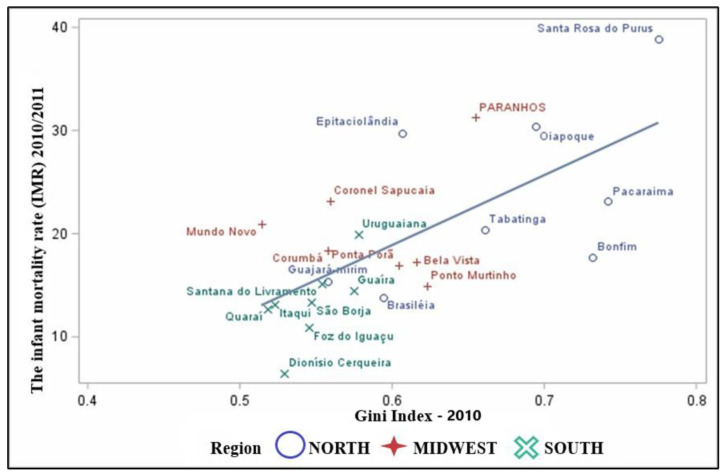
Scatter plot between the averages of IMRs proportional to the Gini Index of the Twin Cities.

**Figure 5 children-09-01662-f005:**
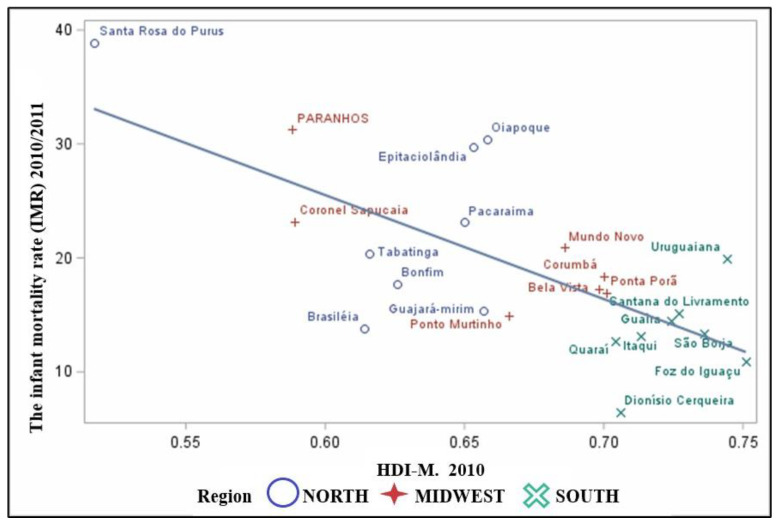
Scatter plot between the means between the averages of the IMRs proportional to the HDI-M.

## Data Availability

The data presented in this study are available in Appendix A.

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
