# Peer review of "Changes in the Infant Mortality Rate in Twin Towns of Brazil: An Ecological Study"

_children, 2022, doi:10.3390/children9111662_

Round 1

Reviewer 1 Report

This study draws important attention to an oft-neglected subject, that of the population health of international border cities. Using different measures of population health, this study looks to shed light on worse population outcomes at border cities in comparison to the broader state that city lies within, with infant mortality rate the specific outcome marker being assessed.  

Due to challenges with English grammar, the manuscript is difficult to read in places from the Abstract through the Results sections; the Discussion and Conclusion sections read more easily. 

In line 12, it is unclear if the general reduction in mortality that is mentioned refers to general reduction in global mortality, state mortality, or other; it would provide clarity to state this in the manuscript. 

In line 15, it would provide clarity to rephrase the sentence to say something like: "The objective of this study is to assess IMR..."

As a general observation, sometimes "Twin Cities" is capitalized and sometimes it is not, appearing as "twin cities". It would improve readability if it was consistently presented the same way throughout the manuscript, if that is appropriate.

In line 19, the statement "Data were identified by place of occurrence and maternal residence" is confusing. It would help by make the statement more clear by saying something such as: "Data were identified by the city in which the infant death occurred in addition to the mother's primary city of residence." It would help to clarify this phrase in the other places it occurs in the manuscript. 

In line 37, "most often early deaths" should be rephrased within that sentence to make it clear how it relates. 

As a general observation, sometimes "live births" is abbreviate to LB and sometimes it is not; making this consistent would improve readability. 

In line 50, there is an incomplete sentence that should be rephrased: "countries, especially in border towns."

In line 60, the abbreviation "TMI" appears. This is the first time it appears and should be spelled out the first time. 

In Figure 1 and Figure 2, the abbreviations on the y-axis should be spelled out in a legend. 

In line 175, please spell out the unit "NV", as in "21.67 deaths/1000NV..."

For Figure 3, what are the units for the y-axis?

For Figures 4 and 5, please provide an English translation of the y-axis and of the legend. 

Please define the acronym "SUS" where it appears in line 301 if this is the first time is appears in the text. 

The sentence in line 324 starting "directly associated with indicators such as the..." appears to be an incomplete sentence. Please rephrase for clarity. 

In line 358, please define the acronym "ESF" if this is the first time it appears in the text. 

Author Response

Dear Reviewer.

We would like to thank you for your careful review and comments. We are now submitting an improved version of the original manuscript. We went through all the comments and the answers can be seen below: In the manuscript for better visualization, we leave it in accordance with reviewer #1 (highlighted in yellow).

Best regards,

De Oliveira, H.M  and Wendland, E.

Reviewer 2 Report

This is a good and, might I add, very necessary manuscript about the comparative infant mortality rates in Brazil. In truth, I don’t fully understand the concept of “twin cities”, but that is beyond the point.

In order to be very brief, I will only point out what I consider needs an improvement:

-          Rows 42-44 – the authors should rephrase the sentence. Is LB4 a typo?

-          Row 50 – the word “countries” is repeated

-          Row 60 – Is TMI the same as IMR? Please replace, if so

-          Rows 72-78 – why is this paragraph relevant? Does this have some link to the concept of “twin cities”? If so, the authors make a poor job of explaining said concept

-          Figure 1 – A and B are identical charts with identical figures. Please cater to this and replace whichever is actually missing

-          Figure 2 – I understand that some data is missing for the second chart (D), but I would love the figure to be rearranged so both charts begin and end at the same points on the horizontal axis. This would dramatically improve the comparative view

-          Please, translate in English, the legends used for Figures 4 and 5

-          The Discussions section is very well documented and uses strong arguments. For a more synthetic overview, I would sum up the causes listed in rows 284-286 as “neonatal/perinatal causes”, meaning that the bulk of the infant mortality lies in the neonatal period. Also, can “diarrhea and ill-defined deaths” on rows 298-299 be summed up as “infectious causes”?

-          Also, overall (although this aspect will probably be catered during the editing stages), the article is spread over an excessive number of pages, which makes it more difficult to follow.

-          The References should be written in the same manner as the rest of the article and following the guidelines of the journal. I believe that the use of different accents and letters specific to the Portuguese language causes different spacing peculiarities to show, that should be catered for.

Author Response

Dear Reviewer.

We would like to thank you for your careful review and comments.

In the manuscript for better viewing, we leave the corrections according to reviewer #2 (highlighted in green).

Best regards,

De Oliveira, H.M  and Wendland, E.

Round 2

Reviewer 1 Report

No additional suggestions. Nice work.